# Expression of Transcripts in Marmoset Oocytes Retrieved during Follicle Isolation Without Gonadotropin Induction

**DOI:** 10.3390/ijms20051133

**Published:** 2019-03-06

**Authors:** Yoon Young Kim, Byeong-Cheol Kang, Jun Won Yun, Jae Hun Ahn, Yong Jin Kim, Hoon Kim, Zev Rosenwaks, Seung-Yup Ku

**Affiliations:** 1Department of Obstetrics and Gynecology, Seoul National University Hospital, Seoul National University College of Medicine, 28 Yonkeun-dong, Chongno-gu, Seoul 110-744, Korea; yoonykim96@gmail.com (Y.Y.K.); obgyhoon@gmail.com (H.K.); 2Biomedical Research Institute, Seoul National University Hospital, Seoul 03080, Korea; bckang@snu.ac.kr (B.-C.K.); ajphd@snu.ac.kr (J.H.A.); 3Department of Biotechnology, The Catholic University, Bucheon 14662, Korea; jwyun@catholic.ac.kr; 4Department of Obstetrics and Gynecology, Korea University Guro Hospital, Seoul 08308, Korea; zinigo@gmail.com; 5Center for Reproductive Medicine, Weill Cornell Medical College, New York, NY 10021, USA; zrosenw@med.cornell.edu

**Keywords:** oocyte, marmoset, in vitro maturation, microRNA

## Abstract

The in vitro maturation of oocytes is frequently used as an assisted reproductive technique (ART), and has been successfully established in humans and rodents. To overcome the limitations of ART, novel procedures for the in vitro maturation of early follicles are emerging. During the follicle isolation procedure, the unintended rupture of each follicle leads to a release of extra oocytes. Such oocytes, which are obtained during follicle isolation from marmosets, can be used for early maturation studies. Marmoset (*Callithrix jacchus*), which is classified as a new-world monkey, is a novel model that has been employed in reproductive biomedical research, as its reproductive physiology is similar to that of humans in several aspects. The ovaries of female marmosets were collected, and the excess oocytes present during follicle isolation were retrieved without pre-gonadotropin induction. Each oocyte was matured in vitro for 48 h in the presence of various concentrations of human chorionic gonadotropin (hCG) and epidermal growth factor (EGF), and the maturity of oocytes and optimal maturation conditions were evaluated. Each oocyte was individually reverse-transcribed, and the expression of mRNAs and microRNAs (miRs) were analyzed. Concentrations of hCG significantly affected the maturation rate of oocytes [the number of metaphase II (MII) oocytes]. The expression of *BMP15* and *ZP1* was highest when the oocytes were matured using 100 IU/L of hCG without pre-treatment with gonadotropins, and that of *Cja-mir-27a* was highest when cultured with follicle stimulating hormone. These results suggest that these up-regulated miRs affect the maturation of oocytes. Interactions with other protein networks were analyzed, and a strong association of BMP15 and ZP1 with sperm binding receptor (ACR), anti-Müllerian hormone (AMH), and AMH receptor was demonstrated, which is related to the proliferation of granulosa cells. Collectively, on the basis of these results, the authors propose optimal maturation conditions of excess oocytes of marmoset without in vivo gonadotropin treatment, and demonstrated the roles of miRs in early oocyte maturation at the single-cell level in marmosets.

## 1. Introduction

Oocyte in vitro maturation (IVM) is a popular technology in assisted reproductive technology (ART). The process is associated with the developmental competency of oocyte; therefore, the precise optimization of conditions is necessary [1]. Factors, environmental and otherwise, affecting the developmental competency of oocytes have been studied using many species, such as mice and domestic animals; however, such animals have different reproductive characteristics than those of humans [2,3,4,5,6,7]. These animals have estrous cycles and multi-fetal pregnancies, while humans undergo a menstrual cycle and mostly have a single-fetal pregnancy. Therefore, because of these differences in reproductive physiology, there is a need for other animal experimental models that are physiologically close to humans, such as non-human primates (NHP).

The common marmoset (*Callithrix jacchus*) is an emerging NHP model for reproductive biology with the advantage of a short gestation period and a reproductive physiology similar to that of humans [8]. Therefore, ART-related studies using the marmoset model are increasing. Furthermore, many of them use a similar protocol as that of human ovarian stimulation [9]. In addition, limited access to using marmoset has diminished ART-related studies, such as the IVM of oocytes or in vitro follicular maturation (IVFM). The IVFM of early follicles is an alternate option for women showing a poor response to gonadotropin treatment, women in early menopause who barely have gonadotropin-responsive follicles, and cancer survivors with extremely decreased fertility due to loss of ovarian function, in order to achieve pregnancy. Therefore, during the isolation of follicles for IVFM, the use of oocytes from the unintended rupture of early follicles could be another strategy that could be studied and developed using marmoset as a reproductive study model. 

The microRNAs (miRNAs, miRs), which are small RNA families of ~25 bases, regulate gene expression at the post-transcriptional level [10,11] in many developmental processes such as embryonic development and maternal-to-embryonic transition, and have been well-studied [12,13,14]. Furthermore, their association with fertility [15,16] and the developmental competence of oocytes during IVM or IVFM are well-known [17,18,19,20,21]. However, their roles in the reproduction processes in marmoset are only the focus of phylogenetic studies that have focused on the roles of microRNAs, and their roles regarding reproduction in marmoset have been merely studied [22,23,24]. The regulatory roles of miRs in oocyte maturation have been demonstrated in the mouse and bovine models, as well as in humans; however, they have been rarely investigated in marmosets. Due to the increasing use of marmoset as a biomedical research model, generations of knockout marmoset are being actively studied; therefore, multiple ways to use extra oocytes released during unintended follicular rupture could be beneficial to studies in the related fields.

In this study, the authors sought to establish optimal IVM conditions for marmoset oocytes retrieved from the ovary, without treatment with gonadotropins, and analyze the differential expressions of miRs according to IVM conditions to elucidate their roles in oocyte maturation. 

## 2. Results

### 2.1. In Vitro Maturation of Marmoset Oocytes at Various Concentrations of Factors

The experimental scheme of this study is presented in Figure 1. Retrieved oocytes were matured in vitro with any one of the following conditions: (1) human chorionic gonadotropin (hCG) 60 IU/L + FSH 200 mIU/mL in MEM α; (2) hCG 60 IU/L + epidermal growth factor (EGF) 5 mg/mL in MEM α; (3) hCG 100 IU/L + EGF 5 mg/mL in MEM α; (4) hCG 60 IU/L + FSH 200 mIU/mL in Universal IVF media; (5) hCG 60 IU/L + EGF 5 mg/mL in Universal IVF media; and (6) hCG 100 IU/L + EGF 5 mg/mL in Universal IVF media. All of the matured oocytes showed polar bodies and clear morphology (Figure 2A).

Among the conditions employed, the addition of hCG 100 IU/L + EGF 5 mg/mL in Universal IVF media (condition 6) was most effective condition for maturation of the retrieved oocytes. In contrast, in conditions using MEM α (e.g., condition 3), arrested or shrunken oocytes were also observed, as well as mature oocytes. Using these in vitro mature oocytes, a fertilization process was attempted and interestingly, the formation of embryos was successfully achieved (Figure 2B). Thereafter, the survival and maturation rate were calculated.

### 2.2. Fertilization of in vitro Matured Oocytes

The fertilization of in vitro matured oocytes was achieved using live sperm injection (Figure 3A). Sperms were collected from marmosets by the induction of ejaculation. It was observed that the quality of sperm varied in individual marmosets due to the age factor. Therefore, a pre-screening of sperm quality, based on normal morphology and swimming activity, was performed. Three-years-old marmosets showed good sperm quality, whereas the sperm of male marmosets aged under 30 months demonstrated weak activity and immature morphology.

The rate of fertilization was the highest in the condition using hCG 100 IU/L + EGF 5 mg/mL in Universal IVF medium (condition 6). The accomplished fertilization was confirmed by the presence of two polar nuclei (2PN) and the development of embryos into the four-cell stage was confirmed (Figure 3B). Development into the four-cell stage was better achieved using condition 6; however, 10% of the fertilized eggs were arrested at the 2PN stage. Development beyond eight cells was achieved after three days of fertilization, and the embryos were cryopreserved for further manipulation.

### 2.3. Localization of Specific Protein in in vitro Matured Oocytes

The expression of alpha tubulin was confirmed in matured oocytes (Figure 4). The zona pellucida (ZP) of marmoset oocyte is thicker than that of mouse; however, it does not interfere in the fluorescence imaging. Tubulin was consistently expressed in oocytes, and the even distribution of the protein was observed. The expression of the protein co-localized with nucleus inside the ZP was demonstrated by merged images using phase-contrast. The expression clearly showed the developmental competency of in vitro matured oocytes.

### 2.4. Differential Expression of miRNA and mRNA in Matured Oocytes

Early development-specific gene expression in the matured oocytes was also analyzed. The expression of ZP1 was up-regulated in condition 6) group (Figure 5A). The expression of BMP15 was also up-regulated when matured using hCG 100 IU/L. The expression of *Oct4* showed differences among the different conditions. *NOBOX* genes were also up-regulated in the condition 6 group (Figure 5B).

The expression of *Cja-miRs* was up-regulated in oocytes matured using Universal IVF medium (conditions 1 and 4). The conditions using Universal IVF medium demonstrated a trend of lower expression compared to MEM α. *Cja-miR-27a* was up-regulated in oocytes matured in the condition 1 group (Figure 6).

### 2.5. Annotation of miRNAs Using Database Set

The correlation of protein interaction was analyzed using KEGG pathway and STRING database (Figure 7). The target genes of up-regulated miRNA were those related to oocyte maturation and estrogen pathways. The up-regulated gene, *ZP*, had a strong correlation with genes such as acrosome reaction regulator (*ACR*) and calcium channel regulator, *CACNA1G*. The correlation between genes was similar to that of mouse. *BMP15* was closely related to growth/differentiation factor (*GDF*) families, BMP families, anti-Müllerian hormone (*AMH*), and AMH receptor II (*AMHR2*), which are related to persistent Müllerian duct syndrome, and those proteins were related to the TGF-beta signaling pathway (KEGG).

## 3. Discussion

The developmental competence of oocytes is one of the most important requirements for ART. Numerous studies focused on the maturation conditions and factors regulating or enhancing developmental competence for a higher yield of in vitro maturation and transition to early stage embryos, e.g., vitamin D [25,26] and inositol [27,28]. Although many of them were conducted using a murine model, the necessity of using an NHP model for pre-clinical evaluation nonetheless remains, due to the difference of reproductive physiology in humans as compared to the mouse model, such as menstrual cycle, the structure of the uterus, implantation process, gestation period, and the number of offspring.

Non-human primates can be classified into old-world and new-world monkeys [29]; the marmoset belongs to the class of new-world monkeys. Their features and application to various biomedical research studies have been mainly conducted in Japan, and recently, the wider usage of marmoset as a model is emerging [23,30,31,32,33,34]. From the perspective of reproductive physiology, non-human primates are closely related to human models; however, due to the high cost of the supply and maintenance of marmosets, a limited number of researchers are using this NHP model. Although ART studies using the marmoset model are reported, studies have been focused on the IVM of oocytes using a human IVF protocol, mainly consisting of inducing maturation using gonadotropins [9,35,36,37,38]. Nevertheless, using NHP as the model for a wider range of female reproductive diseases has arisen [39].

In vitro follicular maturation (IVFM) processes are newly emerging concepts in reproductive medicine due to their roles for women who do not achieve pregnancy; however, their clinical applications have been limited to date. Hence, the evaluation of this technology in NHP should be for wider usage in the future. Our group has previously reported the IVFG process in mouse [17,18,40] and autopsied rhesus monkey models [41]. However, the marmoset is a different strain of monkey, and its ovaries are morphologically similar to those of the rat. Therefore, in this study, the authors envisaged the use of extra oocytes of marmosets from unintended rupture during the follicle isolation procedure. The isolated oocytes were at varied developmental stages due to non-induction using gonadotropins prior to isolation. Ideally, oocytes at various stages of maturation may not be optimal for immediate fertilization; however, a limited supply of marmosets for biomedical research compelled us to undertake the immediate maturation procedures of such oocytes.

The maturation of marmoset oocytes was achieved with various combinations and concentrations of hormones. The progress to maturation was slower in marmosets, as the zona pellucida (ZP) was thicker than that of mice. Although low concentrations of hCG led to the maturation of the oocytes, the most effective condition for the maturation of such non-induced oocytes was a combination of hCG (100 IU/L) with EGF (5 mg/mL) in Universal IVF medium. The concentration ranges of these hormones were relatively higher than those used in mice [42,43]. Some studies have revealed that the concentrations used for marmosets in pre-gonadotropin-induced IVM ranged from 1 to 10 IU with or without a combination of other factors, such as FSH and estradiol [38,44,45,46]. Our results indicated that the immediate IVM of marmoset oocytes from non-gonadotropin-induced ovaries needs a higher concentration of hCG.

ZP is a translucent glycoprotein matrix jelly-like layer of oocytes, which has numerous roles in fertilization, including oocyte activation and acting as receptors for sperms [47,48,49,50]. The *ZP* genes are divided into three classes in mouse and human [51], and their existence is confirmed in the old world monkeys, bonnet monkeys (*Macaca radiata*), [52,53], and new-world monkeys, such as marmosets [54,55]. The expression of *ZP1* was demonstrated to occur at a lower incidence, while the other two genes, *ZP2* and *ZP3*, were expressed according to the different stages of folliculogenesis [54,56]. In this study, the IVM of marmoset oocytes was demonstrated; however, excess oocytes during follicle isolation were not focused as per this experiment. The up-regulation of *BMP15* and *ZP1* indicate that the maturation condition was optimal for marmoset oocytes and that the early maturation of oocytes obtained without ovulation may correlate to the expression of *ZP1*. In addition, the other developmental regulatory genes, *PRDM1* and *NOBOX*, were also up-regulated. Taken together, the authors demonstrated the correlation of *ZP1* to IVM of marmoset oocytes without gonadotropin pre-treatment.

The expression patterns of miRNAs correlated with the maturation rate, which were different from those of mRNA expression. MicroRNAs are well-known regulators of biological phenomena [10], especially in the oocytes of vertebrates [57]. A majority of previous studies were conducted using the murine model, which included oocyte–somatic cell communication [58], the prevention of oocyte apoptosis [59], and the stimulation or inhibition of maturation [60,61,62]. In this study, the authors analyzed the expressions of *Cja-let-7b, Cja-let-7c, Cja-miR-27a*, and *Cja-miR-224*. These miR and *let-7* families are related to oocyte maturation [60], development [63,64], and hormone balance in granulosa cells [65,66].

The evolution and expression features of miRNAs in marmosets have been demonstrated and compared with other monkeys [22,23]. Their roles in male reproduction, such as the morphogenesis of epithelium and tube development, have been studied [24]; however, the distinctive roles of these miRNAs in the female reproductive processes are barely known. The marmoset genome has common and unique features compared to other monkeys, which include rapid changes in the miRNAs expressed in the placenta. The non-synonymous changes of genes involved in reproductive physiology are *GDF9* and *BMP15*. The changes of genes targeted by *let-7* families were distinctive in marmoset, and the roles of miR-*let-7* families in reproduction have been demonstrated in mouse and bovine species [17,63]. In our study, the expression of *let-7b* and *let-7c* was up-regulated in those oocytes cultured using high-concentration hormones, while *Cja-mir-224* expression was decreased. These results were in contrast with previous studies [60], which may be due to the loss of cumulus cells during follicle rupture and using denuded oocytes for immediate IVM.

In conclusion, the authors proposed optimal conditions for the in vitro maturation of marmoset oocytes without pre-induction using gonadotropins, and analyzed the correlation of mRNA and miRNAs related to oocyte development. The maturation needs a higher concentration of hCG, and the miRNAs may regulate the early maturation of marmoset oocytes, as observed in other animal models.

## 4. Materials and Methods

### 4.1. Ethics and Animal Anesthesia

All of the experimental procedures were reviewed and authorized by the Institutional Animal Care and Use Committee (IACUC) of Seoul National University Hospital (SNUH-IACUC No: 15-0032-C1A1(1), 14 January 2016).

Marmosets were imported from Central Institute for Experimental Animals (CIEA) of Japan and maintained in cages in a temperature and light-controlled room (23 ± 3 °C, 40–60% humidity, and 12 h light/dark cycle). They were under veterinary supervision to maintain a healthy status. Animals were sacrificed by intravenous injection of 10 mL of saturated KCl after anesthesia with intravenous ketamine (10 mg/kg), medetomidine (0.04 mg/kg), and vecuronium (0.2 mg/kg).

### 4.2. Isolation of Ovaries and in vitro Maturation of Oocytes

The ovaries (*n* = 4) of female marmosets were removed by laparoscopy. The excised ovaries were transferred in a transport media containing MEM α, 20% fetal bovine serum (FBS), and penicillin/streptomycin (purchased from Invitrogen, Grand Island, NY, USA). Extra oocytes were retrieved by the unintended rupture of follicles during the dissection of the ovaries. Oocytes at the metaphase I stage were selected and matured in vitro for 36 to 48 h in different conditions.

The in vitro maturation media mainly consisted of either MEM α (Invitrogen, Grand Island, NY, USA) or Universal IVF medium (Orgio, Måløv, Denmark), supplemented with different concentrations of human chorionic gonadotropin (hCG, Merck-Serono, Darmstadt, Germany) and epidermal growth factor (EGF, Invitrogen). Maturated oocytes were either utilized for fertilization or subjected to further analyses. The in vitro maturation conditions are summarized in Table 1.

### 4.3. Sperm Collection

Ejaculated viable sperms of marmosets were collected using a 20-mV vibrator applied to the penis of a 3.5-year-old male marmoset. The collected sperms were washed with human tubal fluid (HTF) medium (Orgio) and incubated for 1 h for capacitation. Actively swimming sperms were injected into the prepared, matured oocytes.

### 4.4. Fertilization of in vitro Matured Oocytes

Matured MII oocytes were transferred to a fertilization dish containing SAGE-1 step media (Orgio), and sperms were injected into each drop. The dish was incubated for 1 day, and the formation of two pronucleus (PN) was judged as an occurrence of fertilization, and the development of the zygote into further stages was studied.

### 4.5. Single Cell Reverse Transcription of Oocyte

Each collected oocyte was washed with HBSS (Invitrogen) and mixed with RNA lysis buffer from the Single Cell-to-CT™ kit (Ambion, Waltham, MA, USA) and stored at −20 °C until use for qRT-PCR reaction. Reverse transcription of the single oocyte from marmoset was performed using a Single Cell-to-CT™ kit according to the manufacturer’s instructions. Briefly, the oocyte was mixed with the lysis solution with gentle pipetting and incubated for 5 min at 24 °C. Following the lysis, stop solution and RT mix were added, and the oocytes were incubated using the following conditions: 10 min at 25 °C, 60 min at 42 °C, and 5 min at 85 °C.

### 4.6. Evaluation of miRNAs Using qPCR

The synthesized cDNA were further used as templates for miRNA qPCR or target gene qRT-PCR. Briefly, miRNA PCR was performed using specific primers, and cDNAs were mixed with NCode™ Express SYBR^®^ GreenER™ miRNA qPCR premix (Invitrogen) and amplified using the following conditions: initial incubation for 2 min at 50 °C, followed by 2 min at 95 °C, and then, 40 cycles of 15 s at 95 °C and 60 s at 60 °C. All of the reactions were performed in triplicate, and the Ct value was calculated based on the U6 expression. The specific primers used for qRPCR are shown in Table 2.

### 4.7. qRT-PCR to Evaluate the Levels of Candidate Gene Expression

Synthesized cDNA were mixed with QuantiTect SYBR green PCR premix (Qiagen, Germantown, MD, USA) and specific primers. The amplification conditions were as follows: initial incubation at 95 °C for 15 min, followed by 45 cycles of denaturation at 95 °C for 15 s, annealing at 58 °C for 20 s, and extension at 72 °C for 30 s. All of the reactions were run in triplicate, and the relative gene expression was normalized using the corresponding *GAPDH* expression. The specific primers used for qRT-PCR are shown in Table 3.

### 4.8. Immunostaining of Oocytes

The oocytes were individually fixed with 4% paraformaldehyde (Sigma-Aldrich, Saint Louis, MO, USA) for 20 min at RT. After washing with phosphate-buffered saline (PBS, Sigma-Aldrich), the oocytes were incubated with 3% bovine serum albumin (BSA, Sigma-Aldrich) solution for 12 h to block the non-specific reaction. The oocytes were incubated with primary antibody (mouse alpha tubulin (SantaCruz Biotechnology, SantaCruz, CA, USA)) for 12 h at 4 °C and washed twice with PBS containing 0.05% Tween 20 (PBST). They were further incubated with a secondary antibody (Alexa Fluor 488-labeled donkey anti-mouse IgG (Molecular Probes, Calsbad, CA, USA)) for 1 h at RT and washed twice with PBST. DAPI solution (10 µM, Invitrogen) was added for the staining of the nucleus, and the images were taken using an EVOS-FL microscope (Thermo Scientific, Waltham, MA, USA).

### 4.9. Database Analysis

For analyses of microRNA and target mRNA correlations, Follicle Online (http://mcg.ustc.edu.cn/bsc/follicle), TargetScan (http://www.targetscan.org), miRTarBase (http://mirtarbase.mbc.nctu.edu.tw), STRING (http://string-db.org), DAVID Bioinformatics Resources 6.8 (https://david.ncifcrf.gov), and KEGG pathway enrichment databases were used.

### 4.10. Statistical Analysis

Statistical analysis was performed using SPSS version 21.0 (SPSS Inc., Chicago, IL, USA). Data were expressed as the mean ± standard errors and were analyzed by Student *t*-test. A p value of less than 0.05 was considered to be statistically significant.

## Figures and Tables

**Figure 1 ijms-20-01133-f001:**
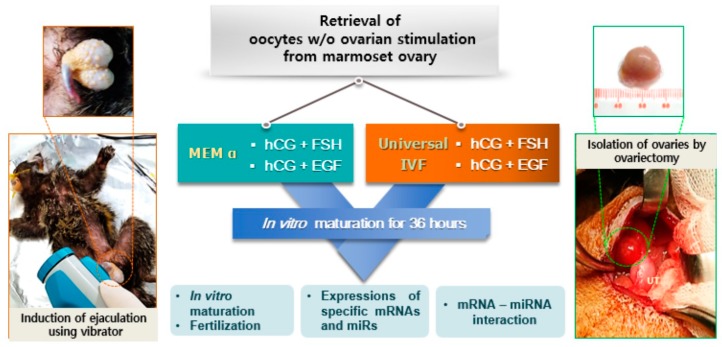
**Experimental flow of oocyte retrieval and in vitro maturation.** Oocytes not pre-treated with gonadotropins were retrieved during follicle isolation. Immediate in vitro maturation of ruptured and retrieved oocytes was conducted in six different conditions; (1) human chorionic gonadotropin (hCG) 60 IU/L + FSH 200 mIU/mL in MEM α, (2) hCG 60 IU/L + epidermal growth factor (EGF) 5 mg/mL in MEM α, (3) hCG 100 IU/L + EGF 5 mg/mL in MEM α, (4) hCG 60 IU/L + FSH 200 mIU/mL in Universal IVF media, (5) hCG 60 IU/L + EGF 5 mg/mL in Universal IVF media, and (6) hCG 100 IU/L + EGF 5 mg/mL in Universal IVF media. Additionally, the collection of sperm by ejaculation using a vibrator from male marmoset is shown (left), and the collection of female ovaries by surgical operation under anesthesia is represented (right).

**Figure 2 ijms-20-01133-f002:**
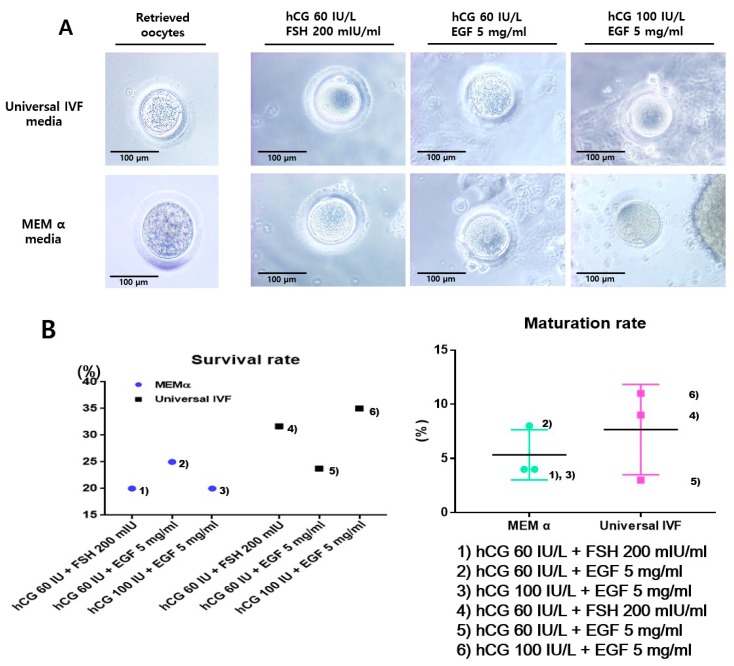
**In vitro maturation of marmoset oocytes using different conditions.** (**A**) Morphological observation of oocytes matured in various conditions. The survival rate was calculated based on the health status of oocytes, and the degenerated oocytes during the maturation process were considered as dead. After 36 h, oocytes with polar bodies were ascertained as matured oocytes. (**B**) Survival and maturation rates of in vitro matured oocytes. Among the maturation conditions, ‘condition (6)’ as mentioned in the methods section was the most efficacious in regard to the survival and maturation of oocytes.

**Figure 3 ijms-20-01133-f003:**
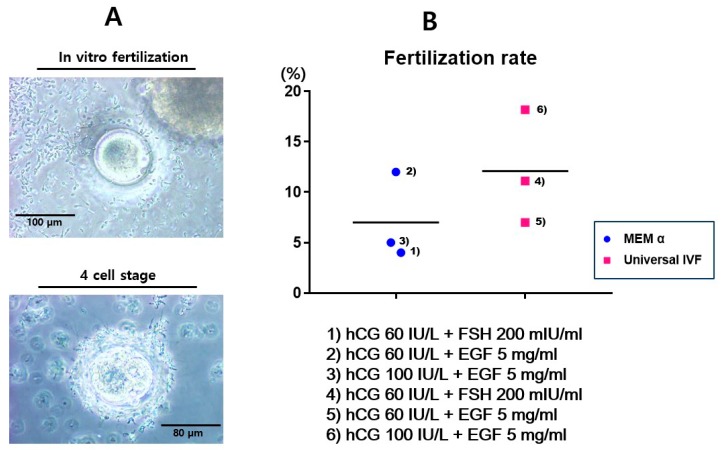
**In vitro fertilization of marmoset oocytes using different conditions.** (**A**) Fertilization and development into the four-cell-stage. In vitro fertilization (IVF) was achieved using viable sperm, and fertilization was confirmed by observation of 2-pronuclei (2PN). In vitro development into the further stage was confirmed after 48 h. (**B**) Fertilization rate using each condition of hormone combinations. The rates were calculated based on the appearance of 2PN after IVF.

**Figure 4 ijms-20-01133-f004:**
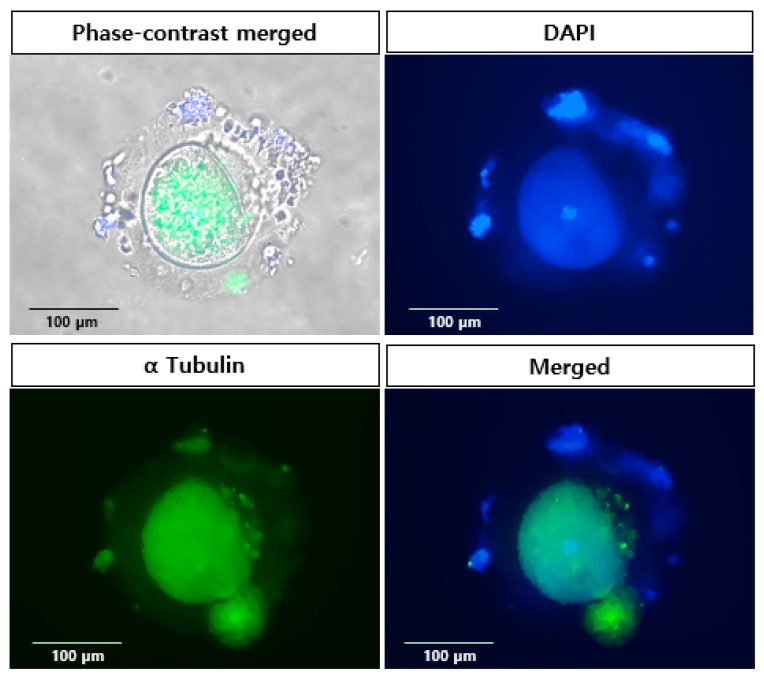
**Localization of specific protein in in vitro matured marmoset oocytes.** Immunofluorescence demonstrates the expression of alpha tubulin. The localization was confirmed using merged images of phase-contrast and fluorescence images. Magnification: ×400.

**Figure 5 ijms-20-01133-f005:**
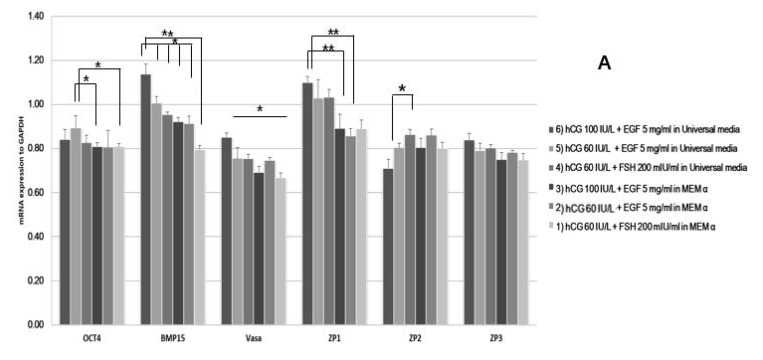
**MicroRNA (mRNA) expression of marmoset single oocytes from different in vitro maturational conditions.** Developmental gene expression of oocytes (**A**) and of primordial germ cells (**B**) was evaluated using qPCR. *BMP15* and *ZP1* were highly up-regulated in the ‘condition 6′ group, and *Novox* was also up-regulated. ** *p* < 0.05.

**Figure 6 ijms-20-01133-f006:**
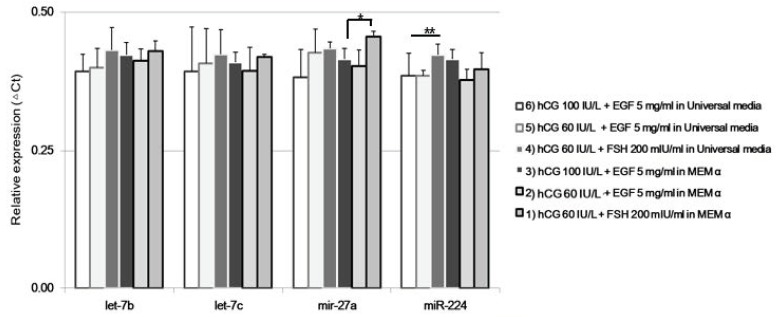
**miRNAs expression of single marmoset oocytes from different in vitro maturation conditions.** The expression of specific miRNAs, *Cja-let-7b, Cja-let-7c, Cja-miR-27a*, and *Cja-miR-224* was analyzed at the single-cell level. * *p* < 0.05; ** *p* < 0.05.

**Figure 7 ijms-20-01133-f007:**
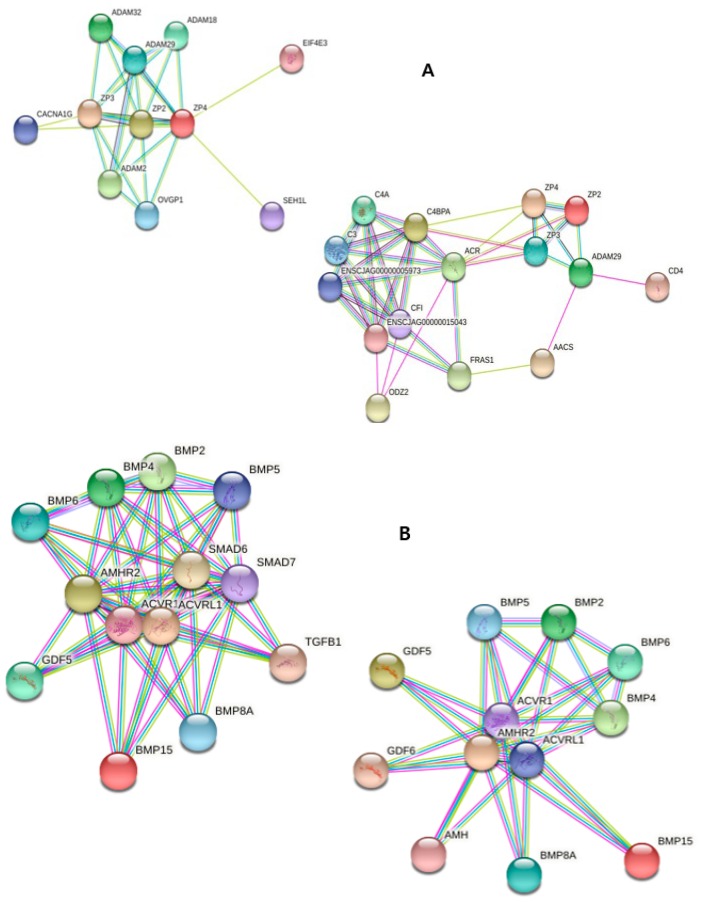
**Predictive interaction of specific proteins in marmoset oocytes.** The interaction of the up-regulated genes with other genes at protein level was analyzed mainly using the STRING database. Comparison within monkey species was conducted; however, the annotation is not demonstrated due to the limitation of marmoset resources.

**Table 1 ijms-20-01133-t001:** Comparison of maturation rate according to media conditions for in vitro maturation (IVM) of marmoset oocytes. MII: metaphase II.

	Media Composition	Number of Matured (MII) Oocytes
MEM α	(1) hCG 60 IU/L + FSH 200 mIU/mL	4
	(2) hCG 60 IU/L + EGF 5 mg/mL	8
	(3) hCG 100 IU/L + EGF 5 mg/mL	4
Universal IVF	(4) hCG 60 IU/L + FSH 200 mIU/mL	9
	(5) hCG 60 IU/L + EGF 5 mg/mL	3
	(6) hCG 100 IU/L + EGF 5 mg/mL	11

**Table 2 ijms-20-01133-t002:** miRNAs specific primer sequences for single-cell profiling of marmoset oocytes.

microRNA	Mature Sequence	Accession ID
*Cja-let-7b*	ugagguaguagguugugugguu	MIMAT0039325
*Cja-let-7c*	ugagguaguagguuguaugguu	MIMAT0049314
*Cja-mir-27a*	uucacaguggcuaaguuccgc	MIMAT0039518
*Cja-miR-224*	caagucacuagugguuccauuu	MIMAT0039393

**Table 3 ijms-20-01133-t003:** Primer sequences for target gene qRT-PCR.

Genes	Forward	Reverse
*Cja_GAPDH*	TGCTGGCGCTGAGTATGTG	AGCCCCAGCCTTCTCCAT
*Cja_OCT4*	GGAACAAAACACGGAGGAGTC	CAGGGTGATCCTCTTCTGCTTC
*Cja_BMP15*	CATTCACTGCGGTACATGCT	TAGTTGGAGATGATGGCGGT
*Cja_VASA*	TGGACATGATGCACCACCAGCA	TGGGCCAAAATTGGCAGGAGAAA
*Cja_ZP1*	CACAGAACAGACCCCCACCTAG	CGCTGGTGGTGTGAGGGAAATG
*Cja_ZP2*	ACTCCCCTCTGTGTTCTGTG	CTGCCTCCTCCCTTGTTT
*Cja_ZP3*	TGTGGCACTCCAAGCCATGC	AGGGCGAGCCACAGGAACCAATG
*Cja_SALL4*	AAGGCAACTTGAAGGTTCACTACA	GATGGCCAGCTTCCTTCCA
*Cja_LIN28A*	GACGTCTTTGTGCACCAGAGTAA	CGGCCTCACCTTCCTTCAA
*Cja_PRDM1*	ATGAAGTTGCCTCCCAGCAA	TTCCTACAGGCACCCTGACT
*Cja_PRDM14*	CGGGGAGAAGCCCTTCAAAT	CTCCTTGTGTGAACGTCGGA
*Cja_DAZL*	GAAGAAGTCGGGCAGTGCTT	AACGAGCAACTTCCCATGAA
*Cja_DPPA3*	GCGGATGGGATCCTTCTGAG	GAGTAGCTTTCTCGGTCTGCT
*Cja_NOBOX*	GAAGACCACTATCCTGACAGTG	TCAGAAGTCAGCAGCATGGGG
*Cja_SCP3*	TGGAAAACACAACAAGATCA	GCTATCTCTTGCTGCTGAGT

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
