# Peer review of "Expression of Transcripts in Marmoset Oocytes Retrieved during Follicle Isolation Without Gonadotropin Induction"

_ijms, 2019, doi:10.3390/ijms20051133_

Round 1

Reviewer 1 Report

No comments.

I recommend to be published as is

Author Response

(please see the attached file)

Manuscript number: ijms-444409

Manuscript title: Expression of transcripts in marmoset oocytes retrieved during follicle isolation without gonadotropin induction

The authors deeply appreciate the constructive and considerate comments of the reviewers. The comments surely strengthened our manuscript. The raised points are updated in the revised manuscript and marked as red throughout the manuscript.

Reviewer 1

1. Manuscript should be further revised by a native English speaker.

è  The manuscript has been revised by native English speakers and the certificate of editing is attached as follows.

2. Was this study registered? I could not find any information about this point.

è  The authors appreciate this critical comment. The study has been reviewed and registered by the IACUC of Seoul National University Hospital. The registration certificate is attached as follows.

3. Authors should discuss, at least briefly, the potential role of Vitamin D status in modulating the oocyte developmental competence, as well as implantation rate. Some interesting articles about the topic are: PMID:29028072; PMID:27740917; PMID: 26943610.

è  The authors appreciate constructive comments. The authors now discussed the role of Vitamin D on the oocyte developmental competence in the Discussion section. Recommended references have been cited accordingly. The newly phased sentences and references read as follows.

Numerous studies focused on the maturation conditions and factors regulating or enhancing developmental competence for higher yield of in vitro maturation and transition to early stage embryos e.g. vitamin D (Lagana et al., 2017; Nandi et al., 2016) and inositol (Vitale et al., 2016; Reyes-Munoz et al., 2018).

Lagana AS, Vitale SG, Ban Frangez H, Vrtacnik-Bokal E, D'Anna R. Vitamin D in human reproduction: the more, the better? An evidence-based critical appraisal. Eur Rev Med Pharmacol Sci. 2017;21(18):4243-51.

Nandi A, Sinha N, Ong E, Sonmez H, Poretsky L. Is there a role for vitamin D in human reproduction? Horm Mol Biol Clin Investig. 2016;25(1):15-28. doi:10.1515/hmbci-2015-0051.

Vitale SG, Rossetti P, Corrado F, Rapisarda AM, La Vignera S, Condorelli RA et al. How to Achieve High-Quality Oocytes? The Key Role of Myo-Inositol and Melatonin. Int J Endocrinol. 2016;2016:4987436. doi:10.1155/2016/4987436.

Reyes-Munoz E, Sathyapalan T, Rossetti P, Shah M, Long M, Buscema M et al. Polycystic Ovary Syndrome: Implication for Drug Metabolism on Assisted Reproductive Techniques-A Literature Review. Adv Ther. 2018;35(11):1805-15. doi:10.1007/s12325-018-0810-1.

4. Several lines of evidence support the key role of inositol-related pathways to orchestrate oocyte development towards quality and, in this way, also reproductive outcomes of IVF procedures. It would be interesting to discuss lights and shadows of recent literature about the topic, referring to: PMID: 27651794; PMID: 30311070.

è  The authors appreciate the constructive comments of the reviewer. We reviewed the recommended literature, discussed and cited in the Discussion section as indicated in the response to point #3. 

Reviewer 2 Report

I read with great interest the Manuscript titled “Expression of transcripts in marmoset oocytes retrieved during follicle isolation without gonadotropin induction” (ijms-444409), which falls within the aim of International Journal of Molecular Sciences.    

In my honest opinion, the topic is interesting enough to attract the readers’ attention. Methodology is accurate and conclusions are supported by the data analysis. Nevertheless, authors should clarify some point and improve the discussion citing relevant and novel key articles about the topic.

Authors should consider the following recommendations:

-       Manuscript should be further revised by a native English speaker.

-       Was this study registered? I could not find any information about this point.

-       Authors should discuss, at least briefly, the potential role of vitamin D status in modulating the oocyte developmental competence, as well as implantation rate. Some interesting articles about the topic are: PMID: 29028072; PMID: 27740917; PMID: 26943610.

-       Several lines of evidence support the key role of inositol-related pathways to orchestrate oocyte development towards quality and, in this way, also reproductive outcomes of IVF procedures. It would be interesting to discuss lights and shadows of recent literature about the topic, referring to: PMID: 27651794; PMID: 30311070.

Author Response

(please see the attached file)

Manuscript number: ijms-444409

Manuscript title: Expression of transcripts in marmoset oocytes retrieved during follicle isolation without gonadotropin induction

The authors deeply appreciate the constructive and considerate comments of the reviewers. The comments surely strengthened our manuscript. The raised points are updated in the revised manuscript and marked as red throughout the manuscript.

Reviewer 1

1. Manuscript should be further revised by a native English speaker.

è  The manuscript has been revised by native English speakers and the certificate of editing is attached as follows.

2. Was this study registered? I could not find any information about this point.

è  The authors appreciate this critical comment. The study has been reviewed and registered by the IACUC of Seoul National University Hospital. The registration certificate is attached as follows.

3. Authors should discuss, at least briefly, the potential role of Vitamin D status in modulating the oocyte developmental competence, as well as implantation rate. Some interesting articles about the topic are: PMID:29028072; PMID:27740917; PMID: 26943610.

è  The authors appreciate constructive comments. The authors now discussed the role of Vitamin D on the oocyte developmental competence in the Discussion section. Recommended references have been cited accordingly. The newly phased sentences and references read as follows.

Numerous studies focused on the maturation conditions and factors regulating or enhancing developmental competence for higher yield of in vitro maturation and transition to early-stage embryos e.g. vitamin D (Lagana et al., 2017; Nandi et al., 2016) and inositol (Vitale et al., 2016; Reyes-Munoz et al., 2018).

Lagana AS, Vitale SG, Ban Frangez H, Vrtacnik-Bokal E, D'Anna R. Vitamin D in human reproduction: the more, the better? An evidence-based critical appraisal. Eur Rev Med Pharmacol Sci. 2017;21(18):4243-51.

Nandi A, Sinha N, Ong E, Sonmez H, Poretsky L. Is there a role for vitamin D in human reproduction? Horm Mol Biol Clin Investig. 2016;25(1):15-28. doi:10.1515/hmbci-2015-0051.

Vitale SG, Rossetti P, Corrado F, Rapisarda AM, La Vignera S, Condorelli RA et al. How to Achieve High-Quality Oocytes? The Key Role of Myo-Inositol and Melatonin. Int J Endocrinol. 2016;2016:4987436. doi:10.1155/2016/4987436.

Reyes-Munoz E, Sathyapalan T, Rossetti P, Shah M, Long M, Buscema M et al. Polycystic Ovary Syndrome: Implication for Drug Metabolism on Assisted Reproductive Techniques-A Literature Review. Adv Ther. 2018;35(11):1805-15. doi:10.1007/s12325-018-0810-1.

4. Several lines of evidence support the key role of inositol-related pathways to orchestrate oocyte development towards quality and, in this way, also reproductive outcomes of IVF procedures. It would be interesting to discuss lights and shadows of recent literature about the topic, referring to: PMID: 27651794; PMID: 30311070.

è  The authors appreciate the constructive comments of the reviewer. We reviewed the recommended literature, discussed and cited in the Discussion section as indicated in the response to point #3.